# Effect of 8 Weeks Aerobic Training and Saffron Supplementation on Inflammation and Metabolism in Middle-Aged Obese Women with Type 2 Diabetes Mellitus

**DOI:** 10.3390/sports10110167

**Published:** 2022-10-30

**Authors:** Ali Rajabi, Mojdeh Khajehlandi, Marefat Siahkuhian, Ali Akbarnejad, Kayvan Khoramipour, Katsuhiko Suzuki

**Affiliations:** 1Department of Exercise Physiology, Faculty of Educational Sciences and Psychology, University of Mohaghegh Ardabili, Ardabil 5619913131, Iran; 2Department of Exercise Physiology, Faculty of Physical Education and Sport Science, University of Tehran, Tehran 141556619, Iran; 3Neuroscience Research Center, Institute of Neuropharmacology, Department of Physiology and Pharmacology, Afzalipour School of Medicine, Kerman University of Medical Sciences, Kerman 7616914115, Iran; 4Student Research Committee, Kerman University of Medical Sciences, Kerman 7619813159, Iran; 5Faculty of Sport Sciences, Waseda University, Tokorozawa 359-1192, Japan

**Keywords:** aerobic training, inflammation, metabolism, saffron, type 2 diabetes, adiponectin, resistin, irisin, pro-inflammatory cytokines, insulin resistance, obesity

## Abstract

Background: This study aimed to investigate the effects of 8-week aerobic training (AT) and saffron supplementation on inflammation and metabolism in middle-aged obese women with type 2 diabetes mellitus (T2DM). Methods: Thirty-two obese women with T2DM were randomly divided into four groups (n = 8 in all groups): saffron + training (ST), placebo + training (PT), saffron supplementation (SS), and placebo (P). The ST and PT groups performed eight weeks of aerobic training (AT) (three sessions/week at 60–75% HRmax). A daily dose of 400 mg saffron powder was consumed by the ST and SS groups for 8 weeks. Blood samples were taken after 12 h of fasting, 48 h before the first AT session, 48 h and two weeks after the last AT session. Results: AT, saffron supplementation, and their combination affected body mass index (BMI), homeostatic model assessment for insulin resistance (HOMA-IR), and serum levels of insulin, adiponectin, interleukin-6 (IL-6), high-density lipoprotein cholesterol (HDL-C), cholesterol, and triglyceride (TG) (*p* < 0.05). However, body weight, body fat percentage, and serum levels of glucose, resistin, tumor necrosis factor-alpha (TNF-α), irisin, and low-density lipoprotein cholesterol (LDL-C) showed significant changes in the ST group only (*p* < 0.05). In addition, a significant difference was seen between all factors in post-training and follow-up in the ST group (*p* < 0.05). Conclusions: Saffron supplementation at a dose of 400 mg/day, when combined with AT, could improve inflammation, metabolism, glycemic status, and lipid profile in T2DM patients, and these changes are sustainable at up to 2 weeks of detraining.

## 1. Introduction

Diabetes mellitus (DM) is a global health problem mainly caused by a sedentary lifestyle, poor dietary habits, and obesity [1,2,3]. Obesity can change adipokine secretion and lead to insulin resistance; a close connection has been reported between obesity and type 2 diabetes mellitus (T2DM) [4]. Adiponectin, the most crucial insulin-sensitizing, anti-inflammatory adipokine, and resistin, a newly discovered adipokine directly related to insulin resistance, are the main adipokines influenced by obesity [5,6]. Adiponectin reduces glucose output from the liver by increasing liver sensitivity to insulin [4,6] and increases glucose uptake in the muscles, thus reducing blood glucose [7]. Similar to adiponectin, resistin regulates glucose homeostasis and acts as a physiological antagonist to insulin action in hepatocytes [5].

Another aspect of obesity and T2DM is the increase in the serum levels of inflammatory cytokines such as interleukin-6 (IL-6) and tumor necrosis factor-alpha (TNF-α) [8]. It has been shown that serum levels of IL-6 and TNF-α are correlated with body mass index (BMI) and obesity [4,9]. IL-6 and TNF-α have an inhibitory effect on glucose transporter type-4 (GLUT-4) expression, and their increase leads to insulin resistance [10,11,12]. Furthermore, IL-6 and TNF-α enhance the survival of inflammatory factors such as resistin, reinforcing insulin resistance [11]. Thus, increased resistin and decreased adiponectin secretion, along with increased pro-inflammatory cytokines, such as IL-6 and TNF- α, are the preliminary properties of T2DM [4,11].

Much like adipose tissue, muscles can secrete metabolically active proteins called myokines. Irisin is a myokine produced by the fibronectin type III domain-containing protein-5 (FNDC5) gene, and its levels decrease in T2DM. Irisin stimulates uncoupling protein-1 (UCP1) production, helping white-to-brown adipose tissue conversion. Considering UCP1′s role in controlling blood glucose, insulin sensitivity, mitochondrial density, and fat metabolism [13], brown adipose tissue is thought to prevent obesity and diabetes [14].

Sedentary behavior and physical inactivity are the main risk factors for T2DM. Thus, aerobic training (AT), the most popular training method in the world [15], could be an essential strategy to counter T2DM [16]. Aerobic training reduces the risk of many conditions. These conditions include obesity, heart disease, high blood pressure, T2DM, metabolic syndrome, stroke, and certain types of cancer [16]. It has been reported that AT could increase the number of GLUT-4 [1]. Thus, it can reduce fasting blood glucose and insulin resistance and reduce inflammation by decreasing serum levels of IL-6, TNF-α, and resistin and increasing serum levels of adiponectin and irisin, which ultimately improve insulin sensitivity [1]. However, some studies have reported no change or increase in the serum levels of inflammatory factors after AT [17].

Supplementation is another strategy to fight T2DM [18]. However, studies have reported that taking chemical supplements may lead to unfavorable side effects, and this has shifted researchers’ attention towards using organic supplements with no or fewer side effects [19]. One of the most popular organic supplements with various medical applications is saffron. The scientific name of saffron is Crocus sativus L. from the Iridaceae family. The most important compounds in saffron are aldehydes (picrocrocin and safranal) and flavonoids (crocetin and crocin) [20]. Research has shown that saffron supplementation improves diabetes by activating the AMP-activated protein kinase (AMPK)/GLUT-4 pathway [21]. Furthermore, it modifies inflammation and oxidative stress [22]. However, studies showed that the interaction of saffron supplementation and combined training has more efficient effects on anti-inflammatory status [23] and pulmonary volume and capacities [24] compared to saffron supplementation or training alone.

Based on the promising effects of both AT and saffron supplementation in improving diabetes-induced inflammation and metabolic disorders, we hypothesized that participating in AT along with saffron supplementation could be even more effective. Therefore, this study aimed to investigate the effect of 8-week AT and saffron supplementation on inflammation and metabolism in middle-aged obese women with T2DM.

## 2. Materials and Methods

This was a quasi-experimental study with repeated measures design: 1: pre-test (48 h before the first AT session), 2: post-test (48 h after the last AT session), and 3: follow-up (2 weeks after the last AT session). Participants consisted of 32 middle-aged obese women with T2DM who lived in Kermanshah Province, Iran. They were selected by convenience sampling method according to the inclusion criteria. The inclusion criteria included no cardiovascular and musculoskeletal disorders, glycosylated hemoglobin lower than 9.9%, no diabetic complications (neuropathy, nephropathy, or retinopathy), no regular AT, no smoking, diabetes history less than 5 years, and a maximum of one type of oral anti-diabetic tablet a day. Participants with blood glucose levels above 120 mg/dL were considered DM. Participants were divided into four groups: saffron + training (ST) (n = 8), placebo + training (PT) (n = 8), saffron supplementation (SS) (n = 8), and placebo (P) (n = 8). The goals and procedure of the study were explained, and written consent was obtained from the participants. During this study, there were no significant changes in the prescription of drugs to control blood glucose or lipids, as reported by the participants’ physicians. The Ethics Committee approved the current study of Mohaghegh Ardabili University (Ethics No: 1605/A. D; date 6 November 2017).

### 2.1. Preparation and Administration of Saffron and Placebo Capsules

Powdered saffron (400 mg) [16] (Food and Drug Organization of Iran Ministry of Health ID: 1021/50 and 111191/50) was placed in capsules and used for two months by the ST and SS groups (once a day at 6 pm). Similar placebo capsules containing 400 mg of wheat flour were prepared for the P and PT groups with the same color and shape as the saffron capsule. All participants were asked to use no drug, except metformin, during the study if possible to control confounding and interfering factors. Major metabolites of saffron were determined using high-performance liquid chromatography.

### 2.2. Controlling Participant Diet

Nutritional data were collected using the 24 h recall diet (two non-holiday days a week and one day at the weekend to determine the mean of nutrient intake). All participants were asked to list all foods and beverages consumed within the last 24 h. All participants completed the questionnaire 30 non-consecutive times (three times a week during the study).

### 2.3. Training Program

An 8 week AT program was performed by the ST and PT groups. Each training session consisted of 10 min warm-up at 55% of target heart rate (THR), 10 min of combined hand-leg movements at 55–60% THR, 30 min of running at 60–75% of THR, and 5 min cool down at 50% THR. Target heart rate was calculated using Karvonen’s formula.
THR = [(MHR − RHR) × %Intensity] + RHR

MHR: heart rate maximum, RHR: resting heart rate.

Participant heart rate was monitored using a polar watch during AT sessions (Polar Electro Oy Professoriate 5, Polar, US6584344, FI-90440 Kempele, Finland). Due to the lack of regular AT and the low physical fitness of participants, the AT program consisted of fast walking in the first two weeks (intensity of 60% THR, duration of 15 to 20 min). The intensity and duration gradually increased each week for the remaining 6 weeks (Table 1). All participants were encouraged to walk and combine hand-leg movements until the end of the session. The P and SS groups did no AT during this period.

### 2.4. Measurement of Variables

A body composition analyzer (In Body version 3.0, made in Seoul, South Korea) was used to measure body weight, Body mass index (BMI), and body fat percentage. Each participant stood on the device without any metal objects and with minimum clothing holding the device electrodes. Measurement was performed at least 3 h after breakfast. Blood sampling was conducted after 10 h of fasting (10 cc venous blood sample from the left radial artery) in 3 stages: pre-test (48 h before the first AT session), post-test (48 h after the last AT session), and follow-up (2 weeks after the last AT session). Serum levels of adiponectin, resistin, IL-6, TNF-α, and irisin were calculated using their optical density at 450 nm by an ELISA reader and using standard curves and applying dilution markers. The ELISA kits were used as follows: adiponectin (Mercodia, Stockholm, Sweden), TNF-α (Orgenium, Helsinki, Finland), IL-6 (Boster, Pleasanton, CA, USA), resistin and irisin (Cusabio, Beijing, China), insulin (Diaplus, North York, ON, Canada), and glucose (Pars Azmun, Tehran, Iran). Serum levels of lipid profile–cholesterol, triglyceride (TG), low-density lipoprotein-cholesterol (LDL-C), and high-density lipoprotein-cholesterol (HDL-C) were also measured enzymatically (Pars Azmun, Tehran, Iran). The insulin resistance index was calculated using homeostatic model assessment for insulin resistance (HOMA-IR) with the following formula [25]:HOMA-IR = fasting blood glucose (mmol/L) × fasting insulin (μu/mL)/22.5

### 2.5. Statistical Analysis

Data were expressed as mean ± standard deviation (SD). Normal data distribution and homogeneity of variances were assessed using the Shapiro–Wilk and Levene tests, respectively. A mixed-model ANOVA with repeated measures across time and group was used to test the main effects and interactions. The Bonferroni post-hoc test was used for checking significant differences among the main effects of each dependent variable. Cohen’s d test was used to estimate the effect size. An effect size of less than 0.2 was considered negligible, between 0.2 and 0.5 small, between 0.5 and 0.8 moderate, and more than 0.8 high. Statistical analysis was performed using SPSS version 22 at a significance level of 5%.

## 3. Results

As shown in Table 2, there was no significant difference between the groups regarding age, energy, and nutrient intake.

The result of the HPLC test is shown in Table 3, in which saffron composition is shown.

Mixed ANOVA showed a significant interaction between groups and time for body weight (F 6, 56 = 11.101, *p* < 0.001), BMI (F 6, 56 = 10.18, *p* < 0.001), and fat percentage (F 6, 56 = 8.98, *p* < 0.001) (Table 4).

The results of mixed ANOVA showed a significant interaction between groups and time for glucose (F 6, 56 = 22.88, *p* < 0.001) (Figure 1), insulin (F 6, 56 = 11.52, *p* < 0.001) (Figure 2), and HOMA-IR (F 6, 56 = 19.34, *p* < 0.001) (Figure 3).

The results of mixed ANOVA showed a significant interaction between groups and time for adiponectin (F 6, 56 = 11.39, *p* < 0.001), resistin (F 6, 56 = 7.55, *p* < 0.001), TNF-α levels (F 6, 56 = 5.87, *p* < 0.001), IL-6 levels (F 6, 56 = 13.86, *p* < 0.001), and irisin (F6, 56 = 9.73, *p* < 0.001) (Table 5).

The results of mixed ANOVA showed that there was a significant interaction between groups and time in LDL (F 6, 56 = 15.42, *p* < 0.001) (Figure 4), HDL-C (F 6, 56 = 24.91, *p* < 0.001) (Figure 5), cholesterol (F 6, 56 = 5.72, *p* < 0.001) (Figure 6), and TG (F 6, 56 = 13.65, *p* < 0.001) (Figure 7).

## 4. Discussion

The present study aimed to investigate the effects of 8-week AT and saffron supplementation on inflammation and metabolism in middle-aged obese women with T2DM. A growing body of evidence has revealed that saffron supplementation could be considered an essential nutraceutical for T2DM treatment due to its hypoglycemic and anti-inflammatory properties [21,22]. Inhibition of oxidative stress, amelioration of ß–cell function, repression of inflammatory pathways, improvement in insulin sensitivity, and stimulation of GLUT-4 expression could be affected by saffron supplementation [21,22,26]. Our results showed that when AT was combined with saffron supplementation, adiponectin, irisin, and HDL-C increased, but resistin, TNF-α, IL-6, body weight, BMI, body fat percentage, LDL-C, cholesterol, blood glucose, insulin, and HOMA-IR decreased. This means that combining AT with saffron supplementation could reduce inflammation, improve metabolism, and ameliorate insulin resistance and body composition.

The present study results indicated that insulin levels and resistance decreased after eight weeks of AT and saffron supplementation, but their combination led to better results. These results are consistent with several studies [15,27] but inconsistent with some others [28,29]. For instance, in a study by Choi et al. [29], where the participants were obese women who performed 45 min/session, 300 kcal/session AT, and 20 min/session, 100 kcal/day strength training, five times a week for three months, a decrease in insulin levels and insulin resistance was not seen. Different diets, the age and sex of participants, and the type of AT protocol may explain this inconsistency [30].

Blood glucose levels only decreased in the ST group. This means that neither AT nor saffron supplementation could affect blood glucose alone. In line with this result, Cauza et al. [31] and Bello et al. [32] did not report significant changes in blood glucose after 16 and 8 weeks of AT, respectively. It has been suggested that AT cannot significantly affect blood glucose in the middle-aged and the elderly [33]. In addition, our participants had a long history of diabetes and a high level of insulin resistance so they may have needed a longer AT period. These two reasons may explain why we could not see significant changes in the PT or SS groups. However, combining AT and saffron supplementation provided sufficient stimulation for upregulating AMPK signaling, which resulted in increasing GLUT-4 translocation into the cell membrane and therefore increased glucose uptake and insulin sensitivity [34].

In line with other studies [35,36], the present study showed that while eight weeks of AT and saffron supplementation increased serum adiponectin levels, combining these interventions led to better results. A study by Rezaeeshirazi [36] showed that AT is more effective in ameliorating insulin resistance in T2D. Yokoyama et al. [37] reported that improvement in insulin sensitivity induced by diet combined with AT (40 min/session, five sessions/week cycling, and 10,000 steps/session walking) for three weeks could not alter adiponectin levels in T2DM patients, which is inconsistent with our results. The reasons for the inconsistency may be the different types of AT and participants’ sex [30] (i.e., female in our study versus both male and female in Yokoyama et al.); high levels of testosterone in men compared to women could inhibit production and secretion of adiponectin [38]. The increase in adiponectin levels in the ST group could be explained by the association of adiponectin with lipid profile (LDL-C, HDL-C, cholesterol, and TG). Adiponectin concentration is directly related to HDL-C levels and inversely related to LDL-C and TG levels [39,40]. This study showed that LDL-C, cholesterol, and TG decreased, and HDL-C increased after combining eight weeks of AT with saffron supplementation.

Furthermore, increased plasma-free fatty acids can stimulate adiponectin secretion. Aerobic training, by increasing lipolysis [39,40] and crocin (one of saffron’s components) by inhibiting pancreatic lipase, which indirectly leads to malabsorption of fat, could lead to an increase in plasma-free fatty acids and stimulate adiponectin secretion [41]. In addition, decreasing insulin resistance might be another reason for increasing adiponectin. Based on this result, we concluded that obesity and T2DM are associated with a decrease in adiponectin, which reduces insulin sensitivity. Conversely, the enhancement of serum levels of adiponectin resulting from AT and saffron supplementation could increase insulin sensitivity through insulin-mimicking, insulin-sensitizing, and anti-inflammatory properties [42,43].

Resistin is another crucial link between insulin resistance and obesity. Our results showed no significant decrease in serum resistin levels after eight weeks of AT and saffron supplementation alone but did show a decrease when they were used in combination. Kadoglou et al. [44] reported that after 16 weeks of AT (i.e., 45–60 min running with 50–85% of vo_2max_, four sessions/week), serum levels of resistin decreased in T2DM patients. However, no significant changes were reported in other studies after AT [45,46]. Different AT types, durations, and intensities might explain this inconsistency [30]. Many studies have shown that resistin is directly related to changes in body fat, body weight, BMI, HOMA-IR, and TG [47], which is in line with our results since all the above factors showed significant changes in the ST group. Another important factor is diet, as it is the primary regulator of resistin. This is why we supervised the participants’ diets, where we observed no significant change in nutrient uptake.

Another insulin resistance influencer is an adipo-myokine, irisin, which has a vital role in energy homeostasis and metabolism [48]. While eight weeks of AT and saffron supplementation could not affect irisin serum levels alone, its levels showed a significant increase in the ST group. This result showed that the isolated effect of AT or saffron supplementation is not enough to increase irisin levels, but the combined effects are needed. However, some studies showed that AT could increase irisin serum levels in diabetic patients [49,50]. Although some studies have suggested a positive correlation between plasma irisin and lactate levels [51], our AT protocol did not have enough load to induce changes in irisin levels.

Our results also showed that while both AT and saffron supplementation could reduce serum IL-6 levels, combining these interventions led to better results. The antioxidant properties of saffron [22] and AT [52] together can reduce reactive oxygen species levels, which play an essential role in the production of pro-inflammatory cytokines [53]. In fact, crocin improves the expression of phosphorylated Akt (p-Akt) and phosphatidylinositol 3-kinase (PI3K). PI3K/Akt signaling plays a significant role in blocking oxidative stress and suppressing the pro-inflammatory response to oxidative stress [54]. The effectiveness of 8 weeks of AT on IL-6 serum levels can also be due to body weight loss, mainly due to the reduced adipose tissue content [55]. IL-6 is secreted by adipose tissue as adipokines, and a decrease in fat mass is associated with a decrease in IL-6 levels [55]. Some studies have reported no significant changes in IL-6 levels after AT [56,57], but others have shown decreased inflammatory and increased anti-inflammatory cytokines in T2DM patients [58].

Another pro-inflammatory cytokine, TNF-α, showed a significant reduction in the ST group only, indicating that it may need a higher amount of AT or saffron to induce isolated changes. In other words, IL-6 seems to be more sensitive to our interventions than TNF-α. The lipid profile, including LDL-C, cholesterol, and TG serum levels, showed a significant reduction in the ST group, and HDL-C showed a significant increase in the ST, SS, and PT groups with no significant difference between them. However, an additive effect was seen for cholesterol and TG highlighting their sensitiveness to AT and saffron. In line with this study, other studies have shown that both saffron and AT [59,60] could improve the lipid profile. Based on previous evidence, saffron supplementation causes an increase in the activity of lecithin-cholesterol acyltransferase (LCAT) [61], which is involved in the regulation of blood lipid metabolism [62]. In addition, regular AT leads to increased daily energy output and fat oxidation and reduces fat mass [63]. As our results show, body weight, BMI, and body fat percentage significantly decreased after eight weeks of AT with saffron supplementation.

The follow-up results (after two weeks) showed that the adaptations disappeared in the PT, SS, and P groups but not in the ST group. It seems that the adaptations in the ST group were high enough to still be detectable two weeks after detraining.

## 5. Conclusions

Saffron supplementation at a dose of 400 mg/day, when combined with AT, could improve inflammation, metabolism, glycemic status, and lipid profile in T2DM patients, and these changes are sustainable at up to 2 weeks of detraining.

## 6. Limitations and Strength

We gave the saffron/placebo capsules to participants and asked them to take them according to the study plan and followed up with them through phone call. This could be a potential limitation in our study, and a future study should ask the participants to take the supplement in front of the researcher. In addition, we collected participants’ nutritional data three days per week, which might have led to a misinterpretation of the participants’ diets. What we consider the strength in our study is using a free-style rather than traditional training method. Furthermore, measuring various factors that affect metabolism and inflammation helped us to better interpret our data, which could be another strength of this study.

## Figures and Tables

**Figure 1 sports-10-00167-f001:**
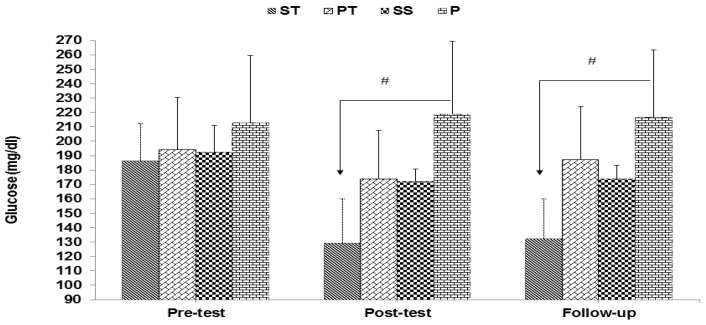
Glucose in the pre-test, post-test, and follow-up (mean ± SD). # indicates a significant decrease in the ST group compared to the P group.

**Figure 2 sports-10-00167-f002:**
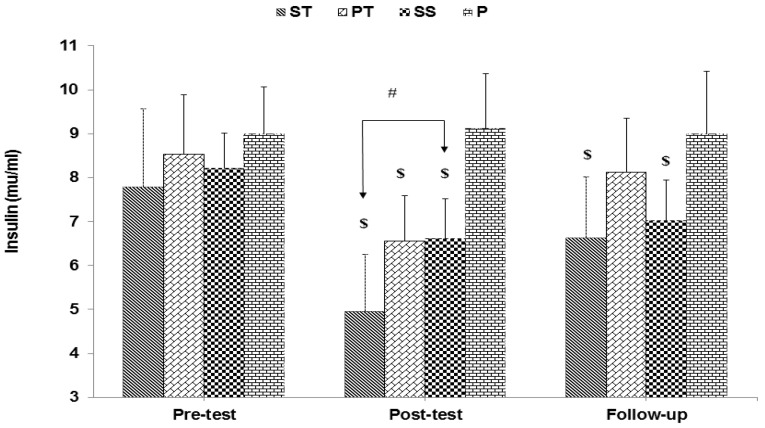
Insulin in the pre-test, post-test, and follow-up (mean ± SD). $ indicates a significant decrease in the ST, PT, and SS groups compared to the P group, and # indicates a significant decrease in the ST group compared to the SS group.

**Figure 3 sports-10-00167-f003:**
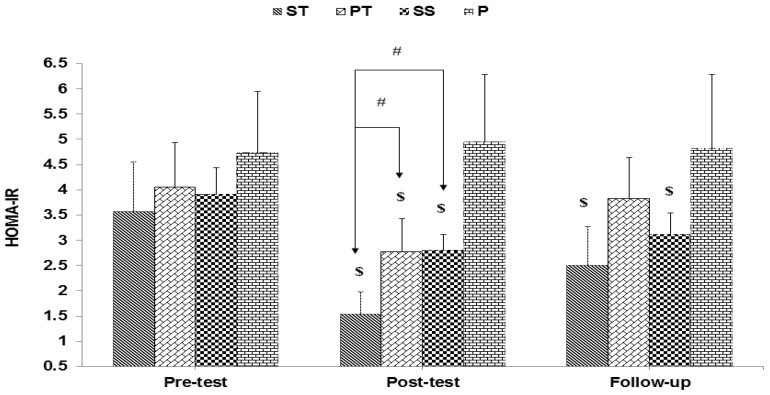
HOMA-IR levels and HOMA-IR. $ indicates a significant decrease in the ST, PT, and SS groups compared to the P group, and # indicates a significant decrease in the ST group compared to the SS group and the PT group.

**Figure 4 sports-10-00167-f004:**
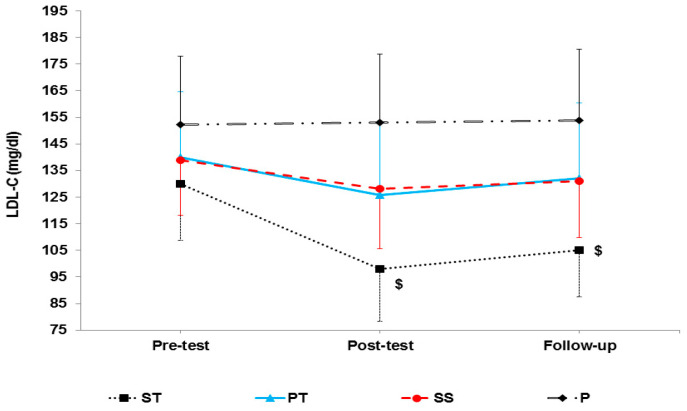
LDL-C levels in the pre-test, post-test, and follow-up. $ indicates a significant decrease in the ST group compared to the P group. $ indicates a significant decrease in the PT group compared to the P group.

**Figure 5 sports-10-00167-f005:**
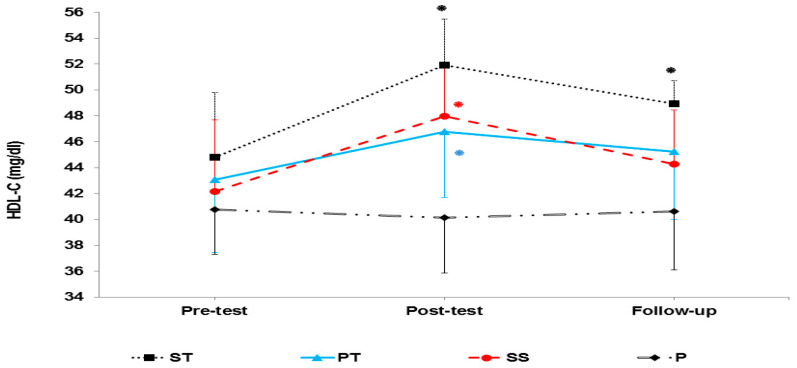
HDL-C levels in the pre-test, post-test, and follow-up. * indicates a significant increase in the ST, PT, and SS groups compared to the P group.

**Figure 6 sports-10-00167-f006:**
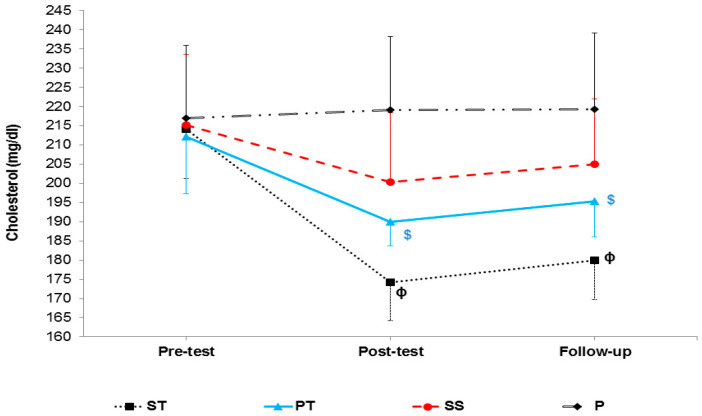
Cholesterol levels in the pre-test, post-test, and follow-up. ɸ indicates a significant decrease in the ST group compared to the SS and P groups. $ indicates a significant decrease in the PT group compared to the P group.

**Figure 7 sports-10-00167-f007:**
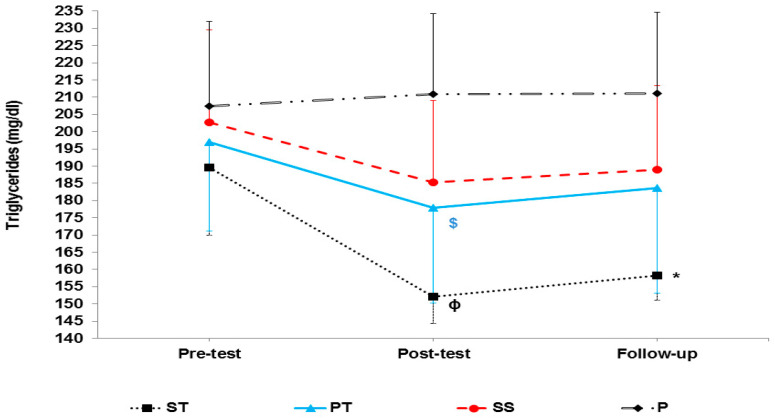
Triglyceride levels in the pre-test, post-test, and follow-up. ɸ indicates a significant decrease in the ST group compared to the SS and P groups, $ indicates a significant decrease in the PT group compared to the P group. * indicates a significant decrease in the ST group compared to the P group in the follow-up.

**Table 1 sports-10-00167-t001:** Aerobic Training Program.

Weeks	Stages of Exercise		Exercise Intensity	Duration per Session (Minute)	Type of Movement
Weeks: 1 and 2: 3 sessions per week	warm-up		%55>	10	Jogging, combined hand-leg movements, stretching movements
	main part	aerobic	%55–60	10	Combined hand-leg movements
		running	%60–65	20	running
	cool-down		%50>	5	Stretching large muscles
Weeks: 3, 4 and 5: 3 sessions per week	warm-up		%55>	10	Jogging, combined hand-leg movements, stretching movements
	main part	aerobic	%55–60	10	Combined hand-leg movements
		running	%65–70	30	running
	cool-down		%50>	5	Stretching large muscles
Weeks: 6, 7 and 8: 3 sessions per week	warm-up		%55>	10	Jogging, combined hand-leg movements, stretching movements
	main part	aerobic	%55–60	10	Combined hand-leg movements
		running	%70–75	40	running
	cool-down		%50>	5	Stretching large muscles

**Table 2 sports-10-00167-t002:** Participants’ age, energy, and nutrient intake (mean ± SD).

Variable		ST	PT	SS	P
Age (year)		51.5 ± 6.16	57.62 ± 6.81	54.12 ± 7.37	56.87 ± 5.11
Disease duration (year)		4.5 ± 3.2	4.6 ± 2.8	3.8 ± 1.7	4.2 ± 3.0
**Nutrient**	Energy (calorie/day)	1682.89 ± 135	1692.36 ± 166	1687.68 ± 149	1705.71 ± 125
	Carbohydrate (g/day)	244.07 ± 62.48	225.93 ± 59.27	223.26 ± 44.76	252.84 ± 38.34
	Protein (g/day)	66.66 ± 14.33	73.53 ± 17.23	78.8 ± 13.39	80.00 ± 16.42
	Fat (g/day)	45.33 ± 12.78	48.33 ± 13.87	44.05 ± 14.11	46.08 ± 11.13
	Fiber (g/day)	12.75 ± 3.75	13.24 ± 4.45	12.92 ± 5.11	14.18 ± 4.21
	Calcium (mg/day)	262.92 ± 179.02	270.18 ± 201.61	268.87 ± 185.23	286.54 ± 178.11
	Vitamin C (mg/day)	55.69 ± 25.12	59.77 ± 26.24	57.26 ± 27.65	62.97 ± 29.44
	Vitamin E (mg/day)	2.5 ± 1.14	2.94 ± 1.46	2.87 ± 1.67	3.41 ± 2.47
	Selenium (mg/day)	42.90 ± 23.26	48.81 ± 21.67	47.89 ± 22.54	53.65 ± 24.11

**Table 3 sports-10-00167-t003:** Determination of the main metabolites of saffron.

Name of the Substance	Picrocrocin	Crocin	Safranal
**Retention time (minute)**	14.8	16.2	26.1
**Saffron metabolites levels (mg/g)**	6.69	6.47	1.17

**Table 4 sports-10-00167-t004:** Participants’ anthropometric properties (Mean ± SD).

Variables	Groups	Pre-Test	Post-Test	Follow-up	Cohen’s D
**Body weight (kg)**	ST	80.98 ± 5.01	77.58 ± 6.37 ^ϕ^	78.12 ± 6.38 ^ϕ^	0.593
	PT	81.87 ± 3.30	80.12 ± 3.47	80.73 ± 3.37	0.516
	SS	81.47 ± 6.91	79.56 ± 7.47	80.06 ± 7.62	0.265
	P	86.95 ± 5.90	87.18 ± 6.32	87.00 ± 6.14	0.037
**Body fat (%)**	ST	31.98 ± 3.66	27.43 ± 2.41 ^ϕ^	28.23 ± 2.88 ^ϕ^	1.468
	PT	33.12 ± 2.19	30.62 ± 1.84	31.42 ± 2.42	1.236
	SS	32.97 ± 3.16	30.87 ± 2.48	31.37 ± 3.20	0.739
	P	34.62 ± 3.06	35.02 ± 2.30	34.87 ± 4.01	0.147
**BMI (kg/m^2^)**	ST	30.32 ± 2.42	28.97 ± 2.86 ^$,ϕ^	29.18 ± 2.88 ^ϕ^	0.509
	PT	31.15 ± 1.50	30.40 ± 1.49 ^#^	30.63 ± 1.31	0.501
	SS	30.72 ± 3.56	30.1 ± 3.75 *	30.18 ± 3.82	0.194
	P	34.03 ± 3.36	34.15 ± 3.57	34.07 ± 3.48	0.034

ST: Training + saffron, PT: Training + placebo, SS: Saffron + supplementation, and P: Placebo. For weight and body fat percentage, ^ϕ^ indicates a significant decrease in the ST group compared to the P group. For BMI ^ϕ^, ^$^, ^#^, and * indicate a significant decrease in the ST group compared to the P group, the ST group compared to the SS group, the PT group compared to the P group, and the SS group compared to the P group, respectively. Cohen’s d test was used to estimate the effect size. BMI: body mass index.

**Table 5 sports-10-00167-t005:** Serum levels of adiponectin, resistin, TNF-α, IL-6, and irisin (mean ± SD).

Variable	Groups	Pre-Test	Post-Test	Follow-up	Cohen’s D
Adiponectin (ng/mL)	ST	13.12 ± 4.18	27.17 ± 7.59 ^&^	23.97 ± 7.02 ^$^	2.293
	PT	11.22 ± 3.81	19.73 ± 4.39 *	15.25 ± 3.77	2.070
	SS	12.03 ± 5.18	18.00 ± 5.92 ^#^	16.18 ± 5.46	1.073
	P	9.15 ± 3.37	9.50 ± 3.42	9.21 ± 3.17	0.102
Resistin (ng/mL)	ST	12.20 ± 2.87	7.02 ± 2.04 ^$^	8.02 ± 2.30 ^$^	2.08
	PT	14.85 ± 3.65	11.93 ± 2.85	14.00 ± 3.07	0.891
	SS	14.07 ± 2.91	12.05 ± 2.61	13.34 ± 2.66	0.730
	P	15.73 ± 5.80	15.85 ± 5.18	14.87 ± 5.67	0.021
TNF-α (pg/mL)	ST	14.05 ± 3.41	9.56 ± 2.79 ^ϕ^	10.25 ± 2.81 ^ϕ^	1.585
	PT	16.65 ± 4.85	15.23 ± 4.69	15.87 ± 4.61	0.299
	SS	16.43 ± 6.64	14.22 ± 4.52	14.62 ± 4.86	0.389
	P	18.10 ± 5.64	18.71 ± 5.58	18.47 ± 5.77	0.108
IL-6 (pg/mL)	ST	9.98 ± 1.94	5.15 ± 1.34 ^$^	7.01 ± 1.51 ^ϕ^	2.897
	PT	11.52 ± 2.14	8.72 ± 2.15 *	9.83 ± 1.64	1.305
	SS	10.93 ± 2.00	9.01 ± 1.52 ^#^	10.55 ± 1.88	1.080
	P	12.43 ± 3.79	12.56 ± 3.71	12.5 ± 4.09	0.034
Irisin (ng/dL)	ST	171.05 ± 26.41	203.92 ± 8.81 ^ϕ^	198.75 ± 7.00 ^ϕ^	1.670
	PT	166.12 ± 21.77	179.72 ± 24.62	176.37 ± 24.31	0.585
	SS	167.07 ± 19.28	177.8 ± 20.14	173.25 ± 23.3	0.544
	P	160.87 ± 17.74	157.15± 18.60	156.98 ± 18.07	0.204

ST: Training + saffron, PT: Training + placebo, SS: Saffron supplementation, and P: Placebo. Serum levels of adiponectin, resistin, TNF-α, IL-6, and irisin in the pre-test, post-test, and follow-up. For adiponectin, ^&^ indicates a significant increase in the ST group compared to the SS and P groups, * indicates a significant increase in the PT group compared to the P group, ^#^ indicates a significant increase in the SS group compared to the P group, and ^$^ indicates a significant increase in the ST group compared to the PT, SS, and P groups in the follow-up. For resistin, ^$^ indicates a significant decrease in the ST group compared to the PT, SS, and P groups. For TNF-α, ^ϕ^ indicates a significant decrease in the ST group compared to the P group. For IL-6, ^$^ indicates a significant decrease in the ST group compared to the PT, SS and P groups, * indicates a significant decrease in the PT group compared to the P group, ^#^ indicates a significant decrease in the SS group compared to the P group, and ^ϕ^ indicates a significant increase in the ST group compared to the P group in the follow-up. For irisin, ^ϕ^ indicates a significant increase in the ST group compared to the P group.

## Data Availability

https://uma.ac.ir/index.php?slc_lang=en&sid=1 (accessed on 5 May 2020).

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
