# Peer review of "Effect of 8 Weeks Aerobic Training and Saffron Supplementation on Inflammation and Metabolism in Middle-Aged Obese Women with Type 2 Diabetes Mellitus"

_sports, 2022, doi:10.3390/sports10110167_

Round 1

Reviewer 1 Report

General comments

The authors have clearly stated that the purpose of the study was to investigate the effects of 8-week aerobic training and saffron supplementation on inflammation and metabolism in middle-aged obese women with type 2 diabetes mellitus The paper is well-written, easy to follow and adds merit to the vital role of combined active lifestyle and proper nutritional supplementation in metabolic health. Given this approach, this work can enhance future attempts in similar research area. However, I have highlighted a few suggestions and concerns in my specific comments section (below) that need to be addressed before considering whether this work should be published or not

Specific comments

INTRODUCTION

- Lines 71-77: I suggest adding a sentence about the beneficial role of aerobic-based activities in general health according to the latest guidelines by the World Health Organization (1).

- Lines 71-77: A statement about the popularity of aerobic training in various forms around the globe according to the latest report by the American College of Sports Medicine (2), it could be a useful addition.

- Line 78: Add a reference after the statement “Supplementation is another strategy to fight against T2DM.”

- Lines 78-87: Add a few sentences presenting evidence in brief from any other relevant training studies that investigated the efficacy of combined exercise and saffron supplementation.

Suggested References:

1.      Bull FC, Al-Ansari SS, Biddle S, Borodulin K, Bumanat MP, Cardonal G, et al. World Health Organization 2020 guidelines on physical activity and sedentary behaviour. Br J Sports Med 2020; 54(24): 1451-1462.

2.      Kercher VM, Kercher K, Bennion T, Levy P, Alexander C, Amaral PC, et al. 2022 Fitness Trends from Around the Globe. ACSMs Health Fit J 2022; 26(1): 21-37.

MATERIALS AND METHODS

-          Was this study registered as a clinical trial before the commencement of the intervention? If so, you should mention that. Otherwise, explain why you did not register this study in advance at any international database in order to raise the credibility and transparency of your work.

-          Why a dose of 400mg saffron was selected? Justify this selection based on scientific evidence or any other relevant studies.  

-           

DISCUSSION

- Lines 268-281: Add the most recent evidence of the effectiveness of aerobic training in cardiometabolic health parameters among populations with obesity or/and type 2 diabetes (3).

- Add a paragraph presenting potential limitations and strengths of the study.

- At the end of the discussion section and after the limitations paragraph, a separate paragraph titled “Conclusions” should be added underlining the main findings and suggesting future research attempts in this area while highlighting potential practical implications.

Suggested reference:

3. Batrakoulis A, Jamurtas AZ, Metsios GS, Perivoliotis K, Liguori G, Feito K, et al. (2022). Comparative efficacy of five exercise types on cardiometabolic health in overweight and obese adults: a systematic review and network meta-analysis of randomized controlled trials. Circulation: Cardiovascular Quality and Outcomes, 15(6), e008243.

Author Response

pleae read attahed file.

Reviewer 2 Report

Based on the promising effects of both AT and saffron supplementation in improving diabetes-induced inflammation and metabolic disorders, the authors hypothesized that participating in AT along with saffron supplementation could be even more effective. Therefore,this study aimed to investigate the effect of 8-week AT and saffron supplementation on inflammation and metabolism in middle-aged obese women with T2DM. The manuscript turns out to be very interesting and well structured. The methodology is good, the results are well presented and the discussions are appropriate and well structured. I have only a few small comments regarding the introduction.

In the introduction section the authors can considere the following mnuscript:

Messina G et al., Exercise causes muscle GLUT4 transolocation in an insulin-indipendent manner, Biology and MedicineOpen AccessVolume 7, Issue Specialissue32015 Article number 006, DOI:10.4172/0974-8369.1000S3007

Moscatelli F et al., Effects of twelve weeks' aerobic training on motor cortex excitability, Journal of Sports Medicine and Physical Fitness, 2020, 60(10), pp. 1383–1389

Author Response

Please read the attached file. 

Round 2

Reviewer 1 Report

No further comments.